# Potential for future reductions of global GHG and air pollutants from circular waste management systems

Adriana Gómez-Sanabria [1,2 ✉], Gregor Kiesewetter[1], Zbigniew Klimont [1], Wolfgang Schoepp[1] & Helmut Haberl [2]

The rapidly rising generation of municipal solid waste jeopardizes the environment and contributes to climate heating. Based on the Shared Socioeconomic Pathways, we here develop a global systematic approach for evaluating the potentials to reduce emissions of greenhouse gases and air pollutants from the implementation of circular municipal waste management systems. We contrast two sets of global scenarios until 2050, namely baseline and mitigation scenarios, and show that mitigation strategies in the sustainability-oriented scenario yields earlier, and major, co-benefits compared to scenarios in which inequalities are reduced but that are focused solely on technical solutions. The sustainability-oriented scenario leaves 386 Tg $CO_{2eq}$/yr of GHG ($CH_4$ and $CO_2$) to be released while air pollutants from open burning can be eliminated, indicating that this source of ambient air pollution can be entirely eradicated before 2050.

[1] Pollution Management Research Group, Energy, Climate and Environment Program, International Institute for Applied Systems Analysis, Laxenburg, Austria.
[2] Institute of Social Ecology, University of Natural Resources and Life Sciences, Vienna, Austria. ✉email: gomezsa@iiasa.ac.at

The quantities of municipal solid waste (MSW) generated globally each year has grown over the last decades, due to population and economic growth, and the consequent changes in production and consumption patterns[1,2]. Estimates suggest that the world population generated 1.9 Gt/yr of MSW in 2015 and is expected to generate about 3.5 Gt/yr of MSW in 2050[3]. High-income countries (World Bank income classification) generate more waste per capita per year than low-income countries: they are responsible for 34% of the amount of MSW generated each year, even though they account for just 16% of the global population[4]. The absence of suitable treatment facilities to cope with the large quantities of MSW will result in various environmental and health impacts[5]. High-income countries can deploy policies and instruments to cope with the rising MSW flows and hence they can potentially have cleaner and better-organized waste management systems. Examples include the EU Waste Framework Directive 2008/98/EC[6] and the amendment EU Directive 2018/851[7], the EU Landfill Directive 1999/31/EC[8] and the amendment EU Directive 2018/850[9], the EU Directive on packaging and packaging waste 94/62/EC[10] and the amendment EU Directive 2018/852[11], the 3 R's strategy in Japan[12] and the Resource Conservation and Recovery Act 1976[13] in the United States. However, measures focusing solely on increasing re-use and recycling have a marginal impact on reducing waste generation[14]. Although some countries e.g., Japan and Netherlands, have managed to reduce MSW generation, most of them are still not successful in reducing the per capita amounts of MSW generated each year[15].

By contrast, low-income countries often lack suitable management systems, resulting from the shortage of funds, poor planning, poor implementation of law, and lack of technology and expertise[4,16,17]. In addition, the outsourcing of resource-intensive production and waste exports from high-income to low-income countries exacerbates the environmental problems resulting from inadequate waste management systems[18]. Often, open burning, littering and poorly managed landfills are the main ways of waste disposal in low-income countries[4]. Open waste burning results in the release of toxic pollutants and greenhouse gases (GHGs)[19–21]. Litter harms wildlife and ecosystems, especially marine life. Global marine litter is currently recognized as one of the biggest sources of ocean's pollution[22,23]. Decomposition of organic matter in landfills can result in the release of methane ($CH_4$)[24], a GHG that is 28 times more potent per kg emitted than carbon dioxide ($CO_2$) in a 100 year timeframe[25]. In addition to the negative impacts on the environment and climate, these unsustainable practices have well-documented adverse effects on human health[26–28].

Recently, research on waste concentrates on the assessment of the linkages between waste and resource use, climate change, air, and water pollution. Previous work determined that only around 13% of the global MSW generated is recycled and 5.5% composted[4]. Furthermore, it is estimated that the relative contribution of energy from waste and wastewater to the global primary energy could increase from 2% to 9% by 2040 and deliver 64 EJ of energy per year (1 EJ = $10^{18}$ Joules) at the end of this period upon implementation of circular management systems[29]. Work focused on GHG and air pollution suggest that landfills contribute about 15% to global anthropogenic $CH_4$ emissions[30] and show that open burning of MSW is an important contributor to particulate matter and air pollutant emissions[20,31,32], specifically, it contributes 11% to total global particulate matter <2.5 μm ($PM_{2.5}$) emissions and 6–7% to total global black carbon (BC) emissions[31,32]. BC from open burning of waste amounts to 2–10% of global $CO_{2eq}$ emissions[33].

However, studies that comprehensively assess and provide evidence of the potential environmental co-benefits resulting from the implementation of circular MSW management systems are rather scarce. Likewise, to our knowledge, no global analysis exists that considers differences between urban and rural settings and assesses how MSW generation, composition, management and associated environmental burdens might change under alternative but plausible future scenarios.

Here, we provide a method to globally assess the current and future MSW generation and composition in urban and rural areas and associated GHG and air pollutant emissions as well as their implications for ambient $PM_{2.5}$. Our global model uses the five Shared Socioeconomic Pathways (SSPs) and a scenario consistent with the future macroeconomic and population pathways of the IEA's World Economic Outlook 2018[34] as activity drivers. Two variant scenarios are developed for each of the six future socio-economic pathways; a 'Baseline - CLE' and a 'Maximum Technically Feasible Reduction – MFR', in which circular municipal waste management systems are implemented globally. We show that the adoption of mitigation strategies in the sustainability-oriented scenario yields earlier, and major, co-benefits compared to scenarios in which inequalities are reduced but that are focused solely on technical solutions. The sustainability-oriented scenario leaves 386 Tg $CO_{2eq}$/yr of GHG ($CH_4$ and $CO_2$) to be released while air pollutants from open burning can be eliminated, indicating that this source of ambient air pollution can be entirely eradicated before 2050. Our detailed representation of the MSW sector and associated emissions and mitigation potentials can be used as input to Integrated Assessments Models (IAMs) applied to develop emission scenarios for the IPCC, support regional and local scale air pollution studies, and inform local and national governments about the likely developments, environmental consequences, and mitigation opportunities in the MSW sector.

## Results

**Scenarios of MSW generation until 2050.** Variations in socio-economic assumptions underlying each of the SSPs lead to significant differences in future MSW flows (Fig. 1). We estimate that the lowest quantities of MSW generation are expected in SSP3 and SSP4 due to slow economic growth and large inequalities between

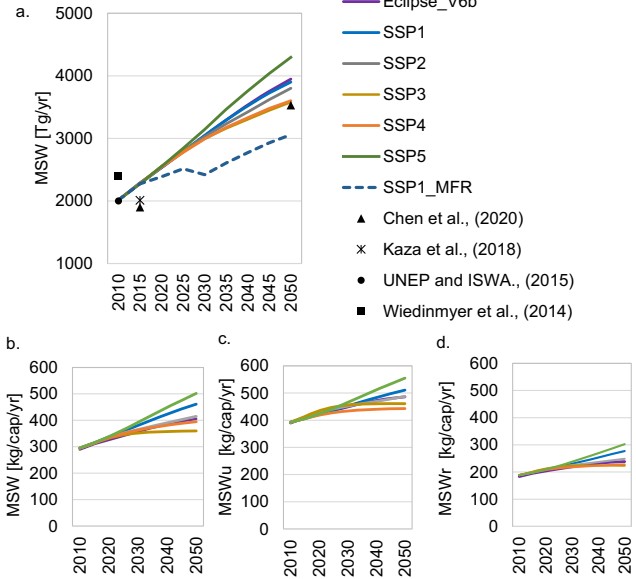

**Fig. 1 Global total and per capita municipal solid waste generation.**
**a** Global total municipal solid waste generation (MSW)[3,4,20,44]. **b** Global MSW generation per capita. **c** Global urban (MSWu) generation per capita. **d** Global rural (MSWr) generation per capita.

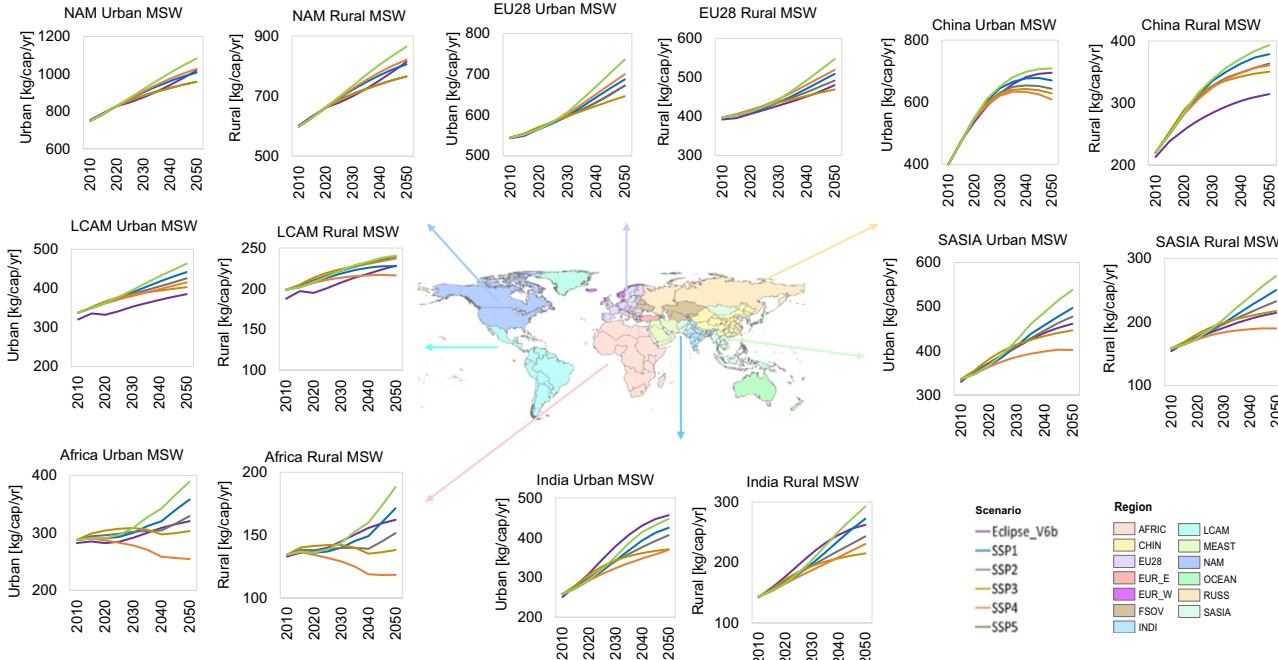

**Fig. 2 Municipal solid waste generation rates in urban and rural areas.** Variations in socioeconomic assumptions result in different municipal solid waste (MSW) trajectories. High-income regions such as North America (NAM) and EU28 are projected to have the highest MSW per capita and year, independent of the underlying socio-economic pathway. However, the different pathway trajectories have a strong influence on MSW per capita generation in low, and middle-income regions such as Africa, Latin America (LCAM) and South Asia (SASIA).

regions. Lower purchasing power in low-income regions reduces the acquisition of durable and non-durable goods thereby reducing MSW generated. By contrast, the highest MSW generation is expected in the SSP5 concomitant with the steep rise in both income and urbanization rates. Our calculations suggest that 4296 Tg/yr of MSW are expected to be generated in 2050 in this scenario. In a sustainability-oriented scenario such as SSP1, MSW generation in 2050 is expected to be 10% lower than in SSP5. However, when improving SSP1 through the adoption of measures targeted at reducing food and plastic waste (SSP1_MFR), it will be possible to reduce MSW generation in 2050 by additional 20% compared to SSP5. Our estimates suggest that urban areas are currently responsible for 70% of the global MSW generated. In 2050, urban areas are expected to generate 80% of the total MSW, whereas the share of rural areas is expected to fall to 20%. This implies that MSW per capita and year is expected to be 50% lower in rural than in urban areas. In general, rural per capita MSW generation is much lower than that in urban areas due to their smaller purchasing power. However, in high-income countries, these differences between urban and rural areas shrink over time.

Figure 2 depicts MSW per capita in urban and rural areas for selected regions. North America (NAM), Europe, Russia, and Oceania are likely to continue having the highest average per capita MSW generation rates in both urban and rural areas during the whole period. However, the calculations reveal relevant differences between and within regions. By 2050, urban NAM is expected to generate in average 1008 (1017–1082) kg/cap/yr of MSW while rural NAM will generate 806 (814–864) kg/cap/yr of MSW. These MSW generation rates are about 35% higher than those estimated for the EU28 and 45% higher than those expected for Oceania. Our estimates also indicate that in a world following the SSP5, urban and rural China will generate between 31 and 36% more MSW per capita and year in 2050 than in 2015. The reason is the stronger economic growth projected in China over the next decade[35]. In 2050, India is expected to generate between 16 and 25% less MSW per capita in urban and

rural areas, respectively, compared to China for the same scenario. The lowest growth on MSW per capita in both urban and rural areas is expected in the SSP3 and SSP4. In general, Africa is projected to continue having the lowest MSW generation across scenarios during the whole period. An average of 355 (254–389) kg/cap/yr in urban areas and 155 (118–188) kg/cap/yr in rural areas of MSW is expected to arise in Africa in 2050. Supplementary Fig. 1 displays total, urban, and rural waste generation by region and scenario.

Unfortunately, regions generating the highest amounts of MSW quantities per year have the lowest collection rates and the poorest MSW management systems. Average MSW collection rates in Africa, India, SASIA, and China are estimated to be in average of about 50–60%, having urban areas collection rates of ~70% and rural areas ~40%. Moreover, the unsuitable management (i.e., disposed in dumpsites or burned without air pollution controls), of the collected fraction exacerbates the already precarious situation. Based on the detailed MSW activity and management strategies matrix of the GAINS model which comprises eight MSW streams and fourteen treatment technologies[29], our estimates suggest that in 2015, 43% of the global MSW collected ended up either in landfills (13%) that are compacted and/or covered but not meeting environmental standards to prevent leakage[36], in unmanaged landfills without any type of management (hereafter referred as dumpsites) (21%), or was openly burned (9%) either directly at the dumpsites (including unintended fires) or in transfer stations. The remaining 29% of the collected waste was either disposed in sanitary landfills (10%), incinerated (high quality with air pollution controls and energy recovery) (7%), recycled (7%), or composted or anaerobically digested (4%), which is mostly happening in high-income countries. From the uncollected fraction, around 20% is estimated to be scattered MSW with a high probability of eventually reaching water courses and 10% openly burned (Fig. 3). The latter estimates are based on global assessments and detailed country-level studies presented in Table 1 in the "Methods" section.

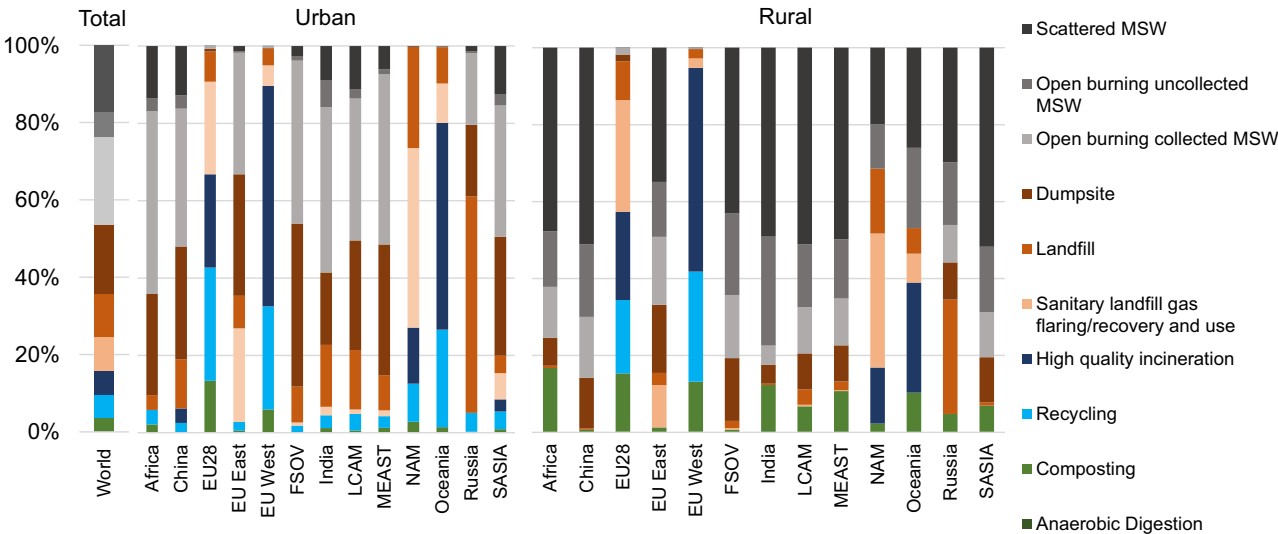

**Fig. 3 Municipal solid waste management in 2015.** Urban areas in low-middle income regions have increased municipal solid waste (MSW) collection rates in last years. However, MSW treatment has not improved at the same pace, hence most of the waste is dumped, scattered or openly burned. Collection rates in rural areas in low-middle income regions are just about 35–45%. In general, high-income regions have established suitable MSW treatment systems in both urban and rural areas. North America (NAM), East Europe (EUEast), West Europe (EUWest), Latin America (LCAM), South Asia (SASIA), Former Soviet Union (FSOV), Middle East (MEAST).

Despite legislation banning open burning of MSW in most of the countries, our calculations indicate that around 16% of global MSW generated (whereof 55% collected and 45% uncollected), was openly burned, which is equivalent to 380 Tg/yr and 394 Tg/yr in 2010 and 2015, respectively. While in urban areas about 60% of MSW burning occurs either on transfer stations or dumpsites, i.e., in the collected fraction, in rural areas is estimated that about 80% of the burning occurs in the uncollected fraction. Rural areas often lack appropriate MSW management systems and therefore the uncollected waste is usually subject to be dumped, scattered or openly burned[37].

If current MSW management strategies are maintained into the future, the expected quantities of MSW disposed in dumpsites and openly burned would rise proportionally to the increase of MSW quantities. In contrast, in an ideal situation where circular MSW management systems (MFR) are implemented globally, it would be possible to avoid almost all dumping and open burning of MSW in 2050, thereby eliminating the environmental and health burdens associated with current management practices. Circular MSW management systems include restrained landfilling of MSW, increase material recycling rates, technological improvement, and implementation of behavioral measures such as reduction of food and plastic waste generation.

**Anthropogenic emissions to air**. We estimate that MSW handling accounts for 8% (30 Tg/yr) of the global $CH_4$ anthropogenic emissions estimated at 344 Tg/yr in 2015[30]. If current MSW management will prevail into the future, average $CH_4$ emissions will increase by 71% (49–55 Tg $CH_4$) over the amount emitted in 2015, thereby contributing to 13% of the global $CH_4$ anthropogenic emissions estimated at 450 Tg/yr in 2050[30]. Under the current management strategies, China, and NAM were in 2015 the leading $CH_4$ generation regions with an average of 4.9 Tg $CH_4$/yr (4.5–5.2 Tg $CH_4$/yr), followed by LCAM (3.7 Tg $CH_4$/yr) and SASIA (2.8 Tg $CH_4$/ yr). If current conditions are maintained until 2050, then India, Middle East, Africa and SASIA will face the highest growth in $CH_4$ emissions from MSW, with an

increase of about 60% compared to 2015 levels. The expected rise of $CH_4$ emissions in these regions is driven by the projected increase of MSW generation couple with the lack of suitable MSW management systems as scattered MSW, dumpsites and precarious landfills (cover or compacted without leakage controls or gas recovery) are the main options to deal with the MSW generated, thereby increasing $CH_4$ emissions.

$CH_4$ emissions from waste deposited of in landfills today will be generated in future years as it depends on the degradability of the organic matter[24]. MSW generation quantities, composition, and policy adoption at early stages makes a significant difference in the trends of $CH_4$ emissions through the years. In a world implementing circular MSW management systems, the maximum diversion of MSW from dumpsites by 2030 is reached in the sustainability-oriented scenario (SSP1_MFR) with 91% less compared to the baseline. This results from the adoption of MSW reduction measures, speedy implementation of anaerobic digestion to treat organic waste, and the establishment of source-separated MSW collection systems to increase the recycling of materials. Our estimates suggest that it will be possible to virtually eliminate the dumping of waste until 2035 in the sustainability-oriented scenario. The adoption of measures is comparatively slower in scenarios depicting high inequalities between and within countries. Therefore, the diversion of MSW from dumpsites takes more time resulting in higher future $CH_4$ emissions. With the exception of SSP1_MFR, in which $CH_4$ emissions are projected to decrease by 4% in 2030, an increase of about 1–2% is expected to happen in all other MFR scenarios compared to the corresponding CLE. The maximum $CH_4$ emission reduction potential by 2050 will be reached in the SSP1_MFR in which $CH_4$ emissions are expected to decrease by 88% compared to the baseline, thus leaving still 187 Tg CO2eq/yr of $CH_4$ to be released in 2050. Other scenarios are expected to release more $CH_4$, namely, SSP3_MFR will leave 663 Tg CO2eq/yr of $CH_4$ and SSP5_MFR 310 Tg CO2eq/yr of $CH_4$ to be emitted by 2050 which is 50% and 80% lower compared to the respective CLE counterparts (Fig. 4).

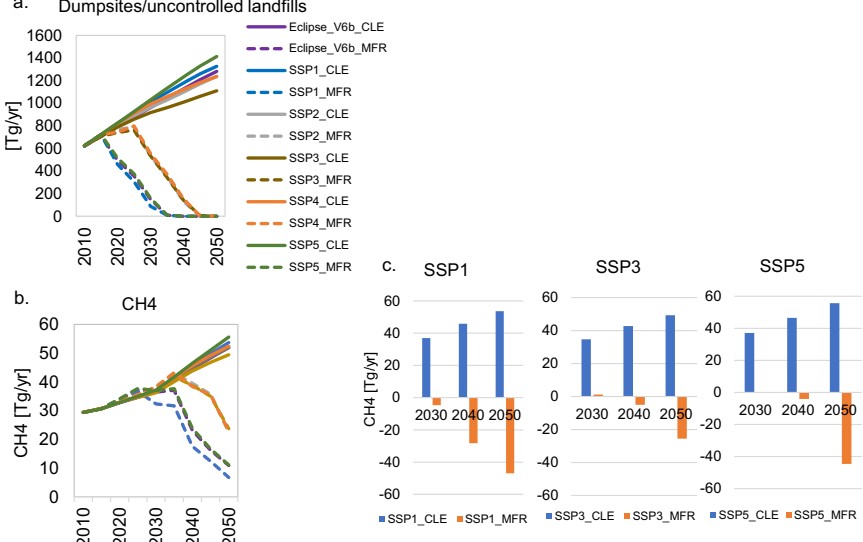

**Fig. 4 Global methane emissions under baseline and mitigation scenarios. a** Municipal solid waste (MSW) in dumpsites, **b** Methane ($CH_4$) emissions and **c** $CH_4$ reduction potentials (negative values in y-axis in panel c refer to avoided emissions). Faster adoption of measures improving MSW systems will result in an early decrease of MSW ending up in dumpsites/uncontrolled landfills and therefore brings quicker reductions of future $CH_4$ emissions from this source. Supplementary Information section 2 presents a detailed analysis of the mitigation (MFR) scenarios.

Emissions of $CO_2$, particulate matter, and air pollutants are released when MSW is burned. Our estimates suggest that globally 150 Tg/yr of $CO_2$ were emitted from MSW combustion in 2015. Please note that we only consider $CO_2$ emissions from fossil carbon while $CO_2$ emissions from biogenic sources are considered net-zero[24], therefore are not reported in this study. We are aware that this simplifying assumption (which is often used) is in many cases not correct[38]. Our results also suggest that the burning of MSW were responsible for 2.5 Tg/yr of $PM_{2.5}$ in 2015. BC emissions are estimated to be 7% and OC 60% of the $PM_{2.5}$ emissions. Overall, $PM_{2.5}$ emissions from MSW account for 8% of the total global anthropogenic $PM_{2.5}$ emissions. Global anthropogenic BC emissions are estimated at 6.0 Tg/yr (GAINS) of which, following our results, 6% are from MSW burning (see Supplementary Table 1 for all pollutant estimates).

Our model indicates that NAM is the largest regional emitter of $CO_2$ from MSW (28 Tg/yr $CO_2$), followed by SASIA (24 Tg/yr $CO_2$), Oceania (19 Tg/yr $CO_2$), Africa (19 Tg/yr $CO_2$) and China (18 Tg/yr $CO_2$). However, the source of these emissions is different. While in NAM and Oceania the main source of emissions is MSW incineration (high-quality incineration with energy recovery), in the remaining countries emissions are primarily generated from open burning of MSW. Under the current conditions, future $CO_2$ emissions in the baseline would probably increase proportionally to the quantities of MSW being incinerated and openly burned, resulting in an average emission of 263 Tg/yr $CO_2$ (in the range from 242 in SSP3 to 308 Tg/yr $CO_2$ in SSP5) across SSPs in 2050 (Fig. 5). Recycling of MSW plays a central part when diverting MSW from combustion processes. The adoption of this measure together with the reduction of plastic MSW generation in the SSP1_MFR results in a maximum reduction of $CO_2$ emissions by 26% in 2050 compared to the corresponding baseline (CLE). All other MFRs scenario families bring $CO_2$ emission increases in the range from 20–25%.

Our calculations also indicate that SASIA plus India, China, Africa, and LCAM emitted 89% of the particulate matter and air pollutants from MSW in 2015. India and China contributed about 50%, Africa 21% and LCAM the remaining 18%. Although

open burning of MSW occurs in the collected and uncollected fraction in both urban and rural areas, most of the emissions come from MSW collected in urban areas. For example, in Indian cities waste handlers burn waste, despite being aware of the ban, mainly due to lack of infrastructure and to prevent accumulation[39]. Furthermore, with the projected growth of MSW generation, and if the current conditions prevail into the future, then the anticipated global emissions of particulate matter and air pollutants from MSW are expected to nearly double in 2050 for all SSPs. SASIA, India, Africa, China and LCAM are expected to be responsible for 93% of the emissions. Future emissions in the CLE scenarios will increase proportionally to the quantities of MSW open burned. Consequently, the reduction of the fraction of MSW being openly burned translates directly into the same particulate matter and air pollutants emission reduction levels (Fig. 5). In that sense, in the SSP1_MFR, SSP5_MFR and ECLIPSE_V6b_MFR scenarios will be feasible to virtually eliminate open burning and therefore this source of air pollution already in 2030 while in the other scenarios this could potentially happen 10–15 years later.

Regions such as Europe have already included the concept of circular economy in various policies and programs as a strategy to reduce consumption of natural resources by means of reuse and recycling, among other measures[40]. Due to the progression of MSW management systems towards sustainability in these regions, future efforts to continue improving MSW systems, thereby reducing GHG and air pollutant emissions is similar across MFR scenarios (Fig. 6). By contrast, middle- and low-income regions show high variation across scenarios due to firstly, the different socio-economic assumptions underlaying each SSP and secondly, the inferior development of MSW systems. The sustainability-oriented scenario (SSP1_MFR) delivers faster emission reductions in both urban and rural areas compared to the other scenarios (see e.g., India and Africa in Fig. 6). Besides, the slow adoption of circular MSW management systems in the MFR scenarios in which inequalities persist results in high emissions of GHG and air pollutants across the years. Thus, dramatically impacting rural areas. Figures for all regions are presented in Supplementary Data[41].

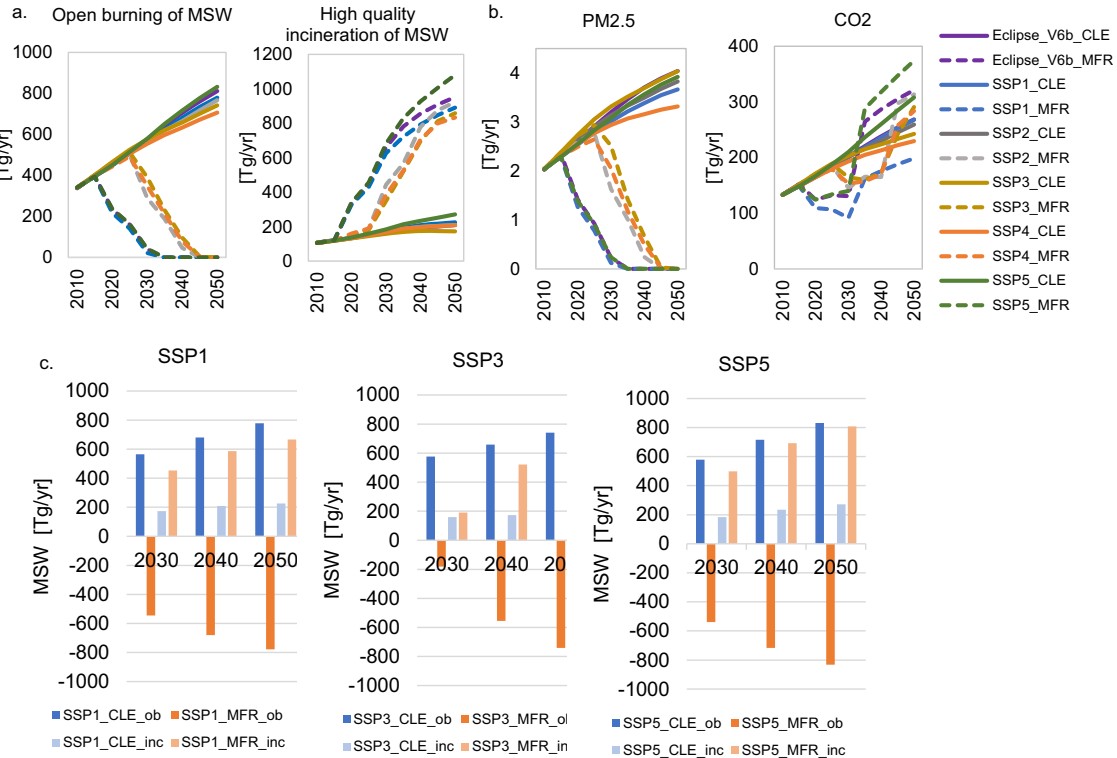

**Fig. 5 Global municipal solid waste incinerated and burned in open fires and related emissions. a** Global amounts of municipal (MSW) incinerated and burned in open fires, **b** related emissions under baseline (CLE) and mitigation (MFR) scenarios and **c** MSW burned in open fires and incinerated (negative values in y-axis refer to the avoided open burning of MSW). Reduction fractions of MSW open burned result in the same reduction percentage of particulate matter and air pollutants. An increase of MSW incinerated with energy recovery is expected in all MFR scenarios. The reduction of $CO_2$ emissions results from the decline of MSW openly burned and increase of recycling. Growth in $CO_2$ emissions in the MFRs families after 2030 is driven by the increase of MSW generation and the diversion of refuse MSW from landfills to incineration with energy recovery. The additional decline in $CO_2$ emissions in the SSP1_MFR is the consequence of reducing plastic waste generation. Supplementary Information section 2 presents a detailed analysis of the MFR scenarios.

As emissions from MSW burning contribute significantly to ambient $PM_{2.5}$, particularly since the sources are often low-level and spatially located close to population, the improvement of MSW management will also have benefits in ambient $PM_{2.5}$. To illustrate the possible contributions and mitigation potential from this sector, we here quantify the contribution of MSW to $PM_{2.5}$ levels in different world regions. Calculations follow the approach applied in ref. [40] and are briefly described in the Methods section below. Differences between the scenarios are driven both by emission changes as well as urbanization trends. Concentrations are highest in India and other South Asia and are expected to grow further under CLE following the emission trends. Other developing regions show similar growth trends but lower absolute concentrations. In China, initial increases level off, peaking around 2035 (SSP1,2,3,4) or 2050 (SSP5). In Europe, NAM and Oceania, contributions from MSW burning are much lower since the combustion happens in well-controlled installations and not as open burning. Gradual implementation of better practices and emission controls eventually decreases concentrations to ~zero before 2050 in all MFR cases, although this is achievable at different points in time depending on the SSP storyline.

Here we present a systemic assessment of reduction potentials of GHGs and air pollutants emissions from implementing circular MSW management systems under six future socio-economic development pathways. The assessment includes the development of two scenarios, namely baseline (CLE) and

maximum feasible mitigation potential (MFR) for each of the pathways. The explicit representation of urban and rural MSW generation, composition and management allows for a deeper analysis of future plausible management and emission trends. This study can assist national, regional, and local governments in developing strategies to limit the release of emissions into the environment as well as support assessments of feasibility and progress in achieving the UN Sustainable Development Goals (SDGs).

Our results show that future MSW generation quantities are expected to be between 1.7 and 2 times higher in 2050 compared to current levels in all scenarios. Our results also highlight that urban areas are responsible for about 80% and will continue being responsible for the higher share of MSW generated in the future. The generally high collection rates of MSW in urban areas do not necessarily imply appropriate management. In SASIA, India, China, LCAM and Africa about 80% of the collected MSW is either dumped or openly burned. Furthermore, most of the MSW generated in rural areas is uncollected, and thus ends up being illegally dumped, scattered, or openly burned resulting in several environmental impacts related to air pollution and GHG emissions and other health and environmental impacts out of the scope of this study. Our findings also indicate that in urban areas about 60% of the open burning occurs either on transfer stations or dumpsites, i.e., in the collected fraction, while in rural areas is estimated that about 80% of the burning occurs in the uncollected fraction.

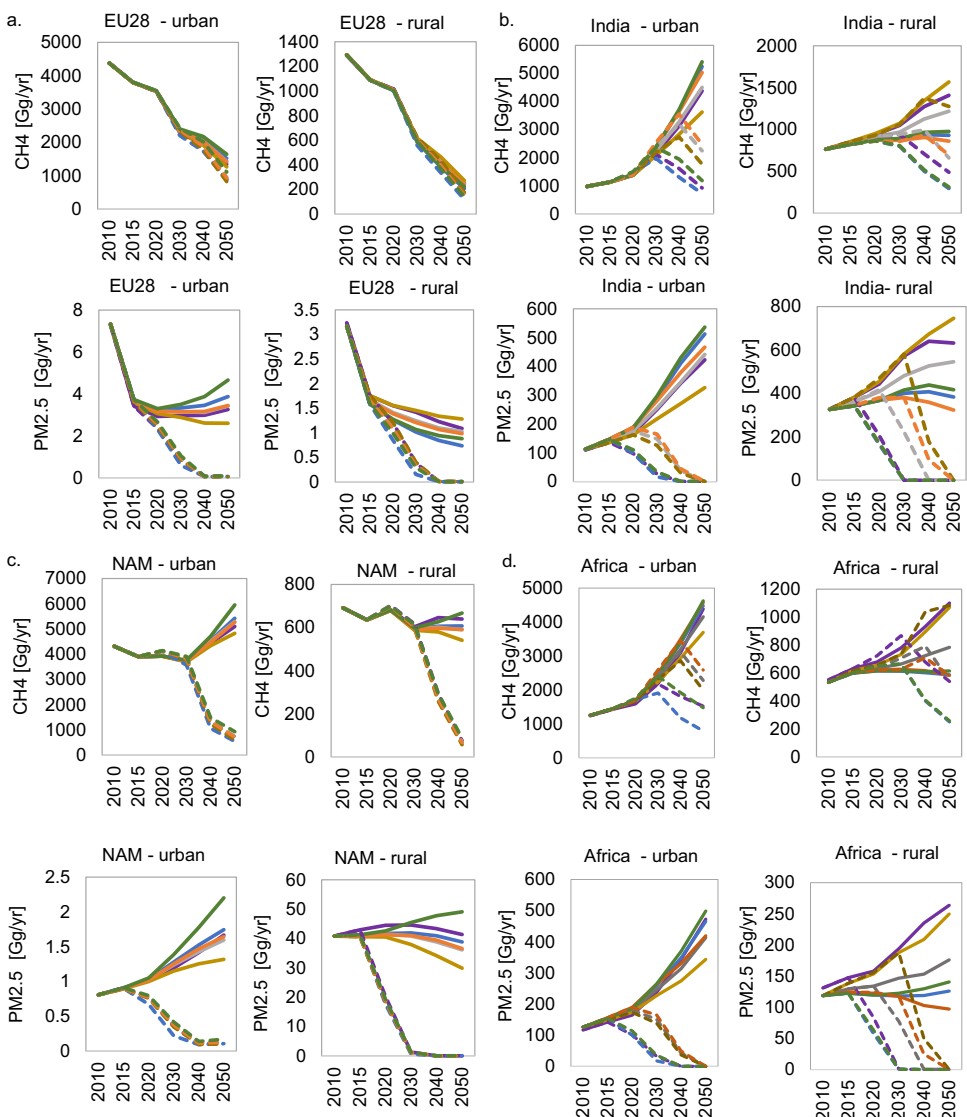

**Fig. 6 Regional emissions of CH₄ and PM₂.₅ from municipal solid waste. a** EU28, **b** India, **c** North America (NAM), **d** Africa. The target of all modeled scenarios is set to reach ~100% of MSW collection and management by 2050. The environmental co-benefits will be obtained at different levels upon the level of socio-economic development and political and institutional arrangements. The different assumptions on policy interventions are then translated into a wide range of future emissions.

In the baseline (CLE), in which current MSW management practices persist without further policy implementation, emissions to air would increase proportionately to the growth in MSW generation. We then developed a set of mitigation scenarios (MFR) to assess the impacts of abatement measures compared to the corresponding baseline (CLE). The common target of our MFR scenarios is to achieve ~100% of MSW collection and treatment by 2050 through the implementation of circular MSW management systems to simultaneously tackle emissions of $CH_4$, $CO_2$, particulate matter, and air pollutants. Co-benefits are obtained at different stages upon the level of socio-economic development and political and institutional arrangements. Evidently, all countries would benefit from reduced MSW generation and improved management in the sustainability-oriented scenario (SSP1_MFR). However, the additional benefits of respective measures are especially relevant for regions generating large MSW quantities and lacking suitable management systems. We show that the environmental co-benefits of avoided MSW generation combined with the speedy

implementation of anaerobic digestion to treat organic waste and the establishment of source-separated MSW collection to increase the recycling of materials (SSP1_MFR) yields major and earlier co-benefits in terms of reducing $CH_4$, particulate matter, and air pollutants. However, more ambitious sustainability-oriented scenarios are crucial to meet the waste-related SDGs, specially the 6.3 target which aims at "By 2030, improve water quality by reducing pollution, eliminating dumping and minimizing release of hazardous chemicals and materials, halving the proportion of untreated wastewater and substantially increasing recycling and safe reuse globally"[42]. We assess that the total eradication of littering and open burning of MSW by 2030 will not be feasible under the current SSP1_MFR. Under this scenario the objective will be reached five years later, i.e., in the year 2035. More ambitious targets and actions are urgently needed to reduce MSW generation and to globally adopt circular MSW systems in order to achieve additional GHG and air pollutant reductions. In addition, the global improvement of MSW systems has the potential to bring progress in other SDGs such as reduction

of food waste (SDG 2), avoiding the release of materials and plastics to water courses (SDG 14), access to energy through energy generation from anaerobic digestion and incineration (SDG 7), reducing GHG emissions to combat climate change and its impacts (SDG 13).

Our analysis also suggests that in 2030, 903 Tg $CO_2eq$ of $CH_4$ (GWP$_{100}$ of 28 $CO_2eq$[25]) will still be released in the SSP1_CLE. Nonetheless, this is 15% lower compared to the $CH_4$ emissions expected in the SSP2_CLE, SSP3_CLE and SSP4_CLE and 13% lower in comparison to the SSP5_CLE and Eclipse_V6b_CLE. Considering that in 2030 high emissions of $CO_2$ from incineration and open burning of MSW would still be released in all MFR families, total GHG emissions ($CH_4$, and $CO_2$) expected in the SSP1_MFR will be about 992 Tg $CO_2eq$/yr which is between 15% and 20% lower compared to the other MFR scenarios. In 2050, SSP1_MFR leaves 386 Tg $CO_2eq$/yr of GHG ($CH_4$ and $CO_2$), to be released. That is 38% lower than the SSP5_MFR and Eclipse_V6b_MFR and 55% lower than the expected emissions in the SSP2_MFR, SSP3_MFR and SSP4_MFR. These variation in emissions can make a substantial difference when considering that the world should stay below 1.5 degrees global warming, i.e., the world can emit as maximum as 10 Pg $CO_2eq$/yr of all GHGs in 2050[43].

The reduction of MSW being openly burned translates into the same reduction level of emissions of particulate matter and air pollutants. Under the development of SSP1_MFR, SSP5_MFR and ECLIPSE_V6b_MFR, the maximum emission reduction potential will be realized in 2030 whereas in the SSP2_MFR will take 5 years more, i.e., in 2040 and for the SSP3_MFR and SSP4_MFR 10 years more, i.e., in 2045. At the same time, MSW combustion contributes to ambient $PM_{2.5}$—in some world regions, this contribution is substantial. Most low-income countries, and particularly those with already high concentrations, show an increasing trend from this source under all SSPs, highlighting the importance of counteracting. The positive message is that mitigation is possible and the MSW contribution to ambient $PM_{2.5}$ can be virtually eliminated by 2050. However, this will not happen by itself.

Comparison to other studies: Our calculations suggest that the world generated 2289 Tg/yr of MSW in 2015. Estimates from other studies vary from 1999[3] to 2010[4] Tg/yr for the same year. Past assessments estimated global MSW generation between 2000[44] to 2400 Tg/yr[20] in 2010. Looking at MSW generation projections, our estimate for the SSP3 and SSP4 in 2050 are similar to the 3539 Tg/yr projected by Chen et al.[3]. Our calculations suggest that although the SSP1 represents a sustainability-oriented pathway, MSW quantities in the baseline are foreseen to reach 3901 Tg/yr in 2050, which is only 10% lower than the expected MSW amounts in the SSP5. Our projection for MSW generation in the SSP2 is 3801 Tg/yr, while ref. [3] estimated a MSW generation of about 3500 Tg/yr in 2050 for the same scenario. However, this estimate is more comparable with our SSP3 and SSP4 projection. The ECLIPSE_V6b_CLE (3948 Tg/yr) is comparable to the SSP1. At the regional level, we find that India is expected to generate about 13% less MSW than China in 2050 across all scenarios. This contrasts findings in ref. [4], in which projected MSW generation in India was about 40% higher than the projection for China in 2050. However, our finding for India is in line with the projection carried out by ref. [45]. Furthermore, the average per capita MSW generation in China is projected to be between 30 and 40% higher than those in India. The fact that estimates for 2010 are lower than those in 2015 and the variability of the results reflect on the one hand, the uncertainty of the data and on the other hand the differences of the methodologies used to derive these numbers. Likewise, our estimate of MSW openly burned is 61% lower than the estimate of ref. [20], who estimated

that 40% or an equivalent of 970 Tg/yr of total MSW generated in 2010 was openly burned (whereof 64% at residential sites and 36% at unmanaged dumpsites) and 57% higher than the estimate of ref. [32], who estimated that about 115 Tg/yr–160 Tg/yr of MSW was openly burned in 2010. Differences in estimated quantities can be attributed to variations in the per capita MSW generation rates adopted, referring partly to different data sources, but also to differences in the methodology used to estimate the fraction of waste openly burned. While the assumption in ref. [20] refers to a fraction recommended in the IPCC (2006) guidelines, we develop our own method which we believe better represents the complexity of the MSW sector e.g., in terms of the urban-rural split and the country/region-specific MSW composition and MSW management pathways (see Methods). The differences of the estimates put a magnifying glass on the urgency to develop national standardized MSW reporting systems, which in addition of being key to governments for the implementation and evaluation of MSW treatment, can serve as part of the monitoring system of GHGs, air pollution and SDGs.

Our estimations indicate that current $CH_4$ emissions from MSW handling account for 8% (30 Tg) of the global $CH_4$ anthropogenic emissions estimated at 344 Tg in 2015[30]. Our estimate is 17% lower than the one estimated by ref. [31] and which has been adopted within the CMIP6 project[45]. It is difficult to assess the level of agreement between both studies as estimates from ref. [31] include MSW and industrial waste while the focus of this study is on MSW and the importance to properly represent the sector for climate and air pollution assessments. However, comparing $CH_4$ emissions from MSW in the Eclipse_V5a[32] to this study, we can see that the estimate in the latter is 6% higher.

Recent global $CO_2$ emissions are assessed at of 39153 Tg/yr in 2015, whereof 130 Tg/yr or 0.33% are generated from waste combustion (including industrial and municipal sources)[31,46]. Reference [20] calculates $CO_2$ emissions from open burning of MSW of 1413 Tg/yr in 2010, estimate that is around 10–15 times higher than that from ref. [31,46] and the one from this study.

In 2010, emissions of $PM_{2.5}$, BC, and OC have been assessed at 6.1, 0.6 and 5.1 Tg, respectively[20]. Our estimates are comparatively lower to those results. In contrast, our results for particulate matter are 60% higher than those from ref. [32]. In both cases the differences are related to the assumed quantities of MSW openly burned. Other studies[31,46] have estimated BC and OC emissions from waste of 0.7 Tg and 4.2 Tg[31], respectively (Supplementary Information section 3 show a comparison of different studies for different pollutants).

## Discussion

Significant potentials exist to reduce GHG, and air pollution provided the implementation of circular MSW management systems. However, the maximum reduction potentials differ between and within regions. Different scenario developments result in similar reduction trajectories in urban and rural areas in high-income regions. In low- and middle-income regions different developments lead to different emission trajectories. Rural areas will be drastically affected in scenarios in which inequalities prevail. The 6.3 target of the SDG 6 can only be achieved through more ambitious sustainability-oriented scenarios that limit MSW generation and improve management. This will require more drastic MSW reduction targets and global adoption of circular MSW management systems before 2030. Similarly, the improvement of MSW systems can directly contribute to the achievement of other SDGs, especially SDG 7, 9, 12, 14, and 15. Our results highlight the importance of acting at various fronts, namely, consumers behavior, technological development, technology transfer, and institutional coordination. For instance, the benefits from reduction of

MSW generation can be jeopardized by social and economic inequalities between and within regions which could restrain the adoption and implementation of measures to improve MSW management systems. Furthermore, for a world focused solely on end-of-pipe solutions will be also beneficial the implementation of policies targeted at reducing MSW generation. The finding is that the development of measures at the consumer side will not bring the expected benefits in terms of emissions reduction if quicker and responsible actions are not taken to bring MSW management systems as an important point in governmental agendas. Certainly, the adoption of circular MSW systems will require actions that incentivize the market of secondary materials and induce consumers and producers to make use of these resources. Regulatory policies associated to assure circularity of products are also necessary. However, it is essential to adopt global binding measures to reduce MSW generation to guarantee success. Finally, we see that the majority of countries have developed some kind of legislation regarding the improvement of MSW management systems, however, the compliance is highly uncertain. A solid system for the reporting of MSW couple with a transparent systematic follow-up of policy enforcement will help to reduce the uncertainty of the estimates as well as will provide clearer insights into the efforts needed by countries to meet their climate, air pollution and SDGs commitments.

## Methods

**General description**. The methodology for developing MSW generation scenarios and associated GHG and air pollutant emissions involves the following five elements: (1) Socioeconomic drivers are taken from the SSP Scenarios for the five SSPs[47] and the IEA-World Energy Outlook 2018[34] and UNDESA[48] for the Eclipse_V6b_CLE (Supplementary Information section 4 presents a short description of the SSPs storylines). (2) The country-specific generation in per capita MSW is driven by expected growth in average per capita income as described in the Supplementary Information section 7 (Supplementary Figs. 2 and 3 show GDP per capita and urbanization rates). (3) Estimation of emissions draw on the methodologies presented in refs. [32,49,50], but are extended to improve source-sector resolution and accommodate for MSW sector-specific information. (4) Implementation of the current legislation for waste management adopted before 2018. (5) Implementation of circular waste management systems are developed in accordance with the EU's waste management hierarchy - Directive 2008/98/EC[6]. The IIASA-GAINS model is used as a framework to carry out this assessment. Supplementary Information Table 6 presents the definition of abbreviations to facilitate the interpretation of terms.

**MSW generation activity and its characteristics**. Here, we have primarily adopted the MSW definition stated in the Directive 851 of the European Parliament and of the council of 30 May 2018 amending Directive 2008/98/EC on waste[7]: "Municipal waste is defined as waste from households and waste from other sources, such as retail, administration, education, health services, accommodation and food services, and other services and activities, which is similar in nature and composition to waste from households. Therefore, municipal waste includes, inter alia, waste from park and garden maintenance, such as leaves, grass and tree clippings, and waste from market and street cleaning services, such as the content of litter containers and sweepings except materials such as sand, rock, mud or dust". However, the definition of MSW generation across countries suffers from inconsistencies thereby introducing higher uncertainties to the estimates. In some cases, amounts reported for MSW generation correspond to the gross quantities of waste collected and in other cases to the MSW quantities left for landfill after quantities separated for treatment have been deducted[51]. In those cases, we have contrasted the maximum sources of information available to adapt the reported data to our core definition.

Current MSW generation quantities, composition, collection rates, and waste management practices are retrieved from several sources, including national official statistics, peer-reviewed literature, and technical reports (see supplement of Gómez-Sanabria et al.[29]. The driver used to project future per capita MSW generation is GDP per capita. This is linked to MSW generation using elasticities estimated following the methodology developed in ref. [50] and further developed in refs. [29,52]. This methodology is further developed in this study (Supplementary Information section 7). Separate elasticities are estimated for groups of countries representing four different average income levels under the assumption that MSW generation and its composition are highly dependent on average national income levels. Furthermore, MSW composition is recalculated based on the estimated income elasticity to per capita food waste generation. MSW composition fractions estimated separately include food, paper, plastic, glass, metal, wood, textile, and other waste. This last fraction includes ordinary mixed waste and may in some cases also include bulk waste.

Quantities and composition of MSW generated by rural and urban population are different. Data on rural waste generation is available for a limited number of countries, when underlying data on rural MSW generation is unavailable, rural waste generation is estimated by applying shares representing the relationship between urban and rural waste generation per capita for different regions using Eqs. (1) and (2). This approach is likely to be an improved version of the one-half rural-urban waste generation ratio used by some studies[4,52] because it captures the differences between regions (Supplementary Information section 8 presents the adopted rural urban rates for different regions).

$$MSW_u = MSW_t * \left( \frac{P_u}{P_u + \left( R_{\left(\frac{r}{u}\right)} * P_r \right)} \right) \quad (1)$$

$$MSW_r = MSW_t - MSW_u \quad (2)$$

where $MSW_t$ is total MSW generated in a country/region (Mt/yr), $MSW_u$ and are MSW generated (Mt/yr) in urban and rural areas, respectively, $R_{\left(\frac{r}{u}\right)}$ represents rural per capita MSW generation as a fraction of the per capita urban MSW generation, and $P_u$ and d $P_r$ is rural are urban and rural population, respectively.

**Open burning of MSW**. In countries without proper implementation of waste legislation, waste mismanagement is aggravated by poor waste separation at the source, low collection rates and low budget allocated to the waste sector[53]. In the absence of reliable waste management systems, dumping and open burning of MSW, either at residential or dumpsites, become the only alternatives to reduce waste- volumes[19,20]. Total MSW openly burned is estimated here as the sum of the fractions of uncollected MSW openly burned and collected MSW openly burned at dumpsites and transfer stations in urban and rural areas. The starting point to derive the quantities of MSW openly burned is the total MSW generated in urban and rural areas. Waste amounts are then split into collected and uncollected waste for urban and rural areas, respectively. Collected waste includes MSW collected by official authorities but also (recyclable) waste collected by the informal sector. Information on collection rates is gathered from sources presented in ref. [29] and complemented from information available in refs. [4,52]. The fraction of uncollected waste is then split into scattered waste or waste openly burned. The fraction of uncollected waste openly burned is assigned based on the information presented in Table 1, considering the current implementation of waste-related legislation, income level, collection rates, and urbanization rate of each region. The fraction of collected MSW openly burned is estimated at 10–20% of the waste ending up in dumpsites, partly due to self-ignition resulting from poor management and partly due to deliberate burning to reduce waste volumes. In addition, a fraction of the collected waste is assumed to be burned at the transfer station or before reaching the disposal site, which is the case in several developing countries[54] Fractions of MSW openly burned, either on the streets or at dumpsites and transfer stations, are dependent on the improvement of the MSW management systems and enforcement of the waste and air pollution legislation. Improvement of waste treatment systems results in reduction of the frequency of MSW openly burned[55]. The quantification of these fractions is however highly uncertain. Literature provides a few different methodologies to estimate the amounts of waste openly burned (Table 1). The IPCC (2006)[24] suggests 0.6 as a representative value for the fraction of total available waste to be burned that is actually openly burned. This assumption is used by Wiedinmyer et al.[20] to estimate GHGs and air pollutants from open burning of waste. Bond et al.[56] assumed lower rates of open burning of waste in rural areas in developing countries based on the statement that most of the waste in rural areas is biodegradable. Table 1 also shows that in many cases the default representative value of the IPCC maybe inadequate for several regions. In general, the quantification of MSW openly burned in region $i$ and year $y$ - $MSW_{(ob)iy}$ is calculated as the sum of MSW openly burned in urban areas $MSW_{(obu)}$ and MSW openly burned in rural areas $MSW_{(obr)}$ applying Eqs. (3, 4, 5).

$$MSW_{(ob)iy} = MSW_{(obu)iy} + MSW_{(obr)iy} \quad (3)$$

Where

$$MSW_{(obu)iy} = [(MSW_{(u)iy} * C_{(u)iy} * (\beta_{0u} + \beta_{1u})) + (MSW_{(u)iy} * (1 - C_{(u)iy}) * \beta_{2u})] \quad (4)$$

$$MSW_{(obr)iy} = [(MSW_{(r)iy} * C_{(r)iy} * (\beta_{0r} + \beta_{1r})) + (MSW_{(r)iy} * (1 - C_{(r)iy}) * \beta_{2r})] \quad (5)$$

Where $MSW_{(u)iy}$ and $MSW_{(r)iy}$ are the total amounts of MSW generated in urban and rural areas in Mt/yr, respectively. $C_{(u)iy}$ and $C_{(r)iy}$ are the MSW collection rates in urban and rural areas, respectively. $\beta_{0u}$ and $\beta_{0r}$ represent the fractions of collected MSW openly burned on transfer stations and $\beta_{1u}$ and $\beta_{1r}$ represent the fractions of collected MSW openly burned at dumpsites in urban and rural areas, respectively. $\beta_{2u}$ and $\beta_{2r}$ are the fractions of uncollected waste openly burned in urban and rural areas, respectively.

**Table 1 Collection of studies quantifying municipal solid waste (MSW) openly burned.**

| Source | Scale | Assumption | Results |
|---|---|---|---|
| Sharma et al.[19] | India | Calculation of waste burned at landfills was based on a study in a landfill in Mumbai using average FRP. Fraction open burning of waste 7– 12% | 68 Tg/yr was open burned in India in 2015 |
| Wang et al.[65] | China | In reference to the limited literature, China's average proportion of open MSW burning is set to 18.0% at residential and dumpsites and 38.0% at landfills. | The proportion of open burning is estimated from 79.8% in 2000 to 57.0% in 2013 |
| Klimont et al.[32] | Global | IPCC guidelines 2006; CEPMEIP, 2002; EAWAG, 2008; Neurath, 2003. Fraction of open burning of waste is 0.5–5% for developed world and 10–20% for developing world. | Global estimation of MSW openly burned is estimated 115 Tg/yr to 160 Tg/yr in 2010 |
| Wiedinmyer et al.[20] | Global | Follows IPCC guidelines 2006 in which 60% of the total waste available to be burned that is actually burned | 970 Tg/yr of waste are globally openly burned. 620 Tg/yr at the residential level and 350 Tg/yr at dumpsites. |
| Hodzic et al.[55] | Mexico City | Assigned percentage of MSW burned according to socioeconomic status. Low and middle-low 60%, mid 30%, mid-high and high 20%. Based on anecdotal evidence with Mexican researchers. | The burned fraction exceeds 4 Gg/day |
| Bond et al.[56] | Global | Fraction of burned waste in urban areas based on United Nations Human Settlement Programme, 2000 | Worldwide 33 Tg/yr, including 14 Tg/yr in Asia and 5 Tg/yr in Africa |

**Emission estimations**. Emissions of non-$CO_2$ greenhouse gases and air pollutants ($E$) by source ($s$) and region ($i$) are calculated in GAINS using Eq. (6)[49]:

$$E_{it} = \sum_{sit} A_{is} * ef_{sm} * Appl_{itsm} \qquad (6)$$

where $A_{is}$ is the activity data in Mt/yr, i.e., the amount of MSW generated before management, $ef_{sm}$ is the emission factor subject to technology $m$, and $Appl_{itsm}$ is the application rate of the technology $m$ to the activity $A_{is}$. The GAINS model matrix comprises fourteen different MSW waste management technologies including different types of source separation, recycling and treatment, different types of solid waste disposal sites and different types of incineration technologies and open burning of waste (Supplementary Information section 9). This extensive characterization of alternative treatment flows allows for a detailed representation of the solid waste management system and its emissions at the national/regional level. Emission factors for $CH_4$ and $CO_2$ are developed according to the 2006 IPCC Guidelines, Volume 5, Chapter 3 and Chapter 5[24]. Emission factors from open burning of waste are adopted as follow: PM emission factors are adopted from ref. [32]. These are 8.75 for $PM_{2.5}$, 5.27 for organic carbon(OC) and 0.65 g/kg for BC. Emission factors for sulfur dioxide ($SO_2$), nitrogen oxides (NOx) and non-methane volatile organic compounds (NMVOC) are adopted from ref. [57,58] and are consistent with ref. [20]. These are 0.225 for $SO_2$, 1.06 for NOx, and 8.46 g/kg for NMVOC. The $PM_{2.5}$ concentrations are obtained using the annual $PM_{2.5}$ emissions applying a simplified version of the atmospheric calculation in the GAINS model[59]. Those estimates build on a linearized representation of full atmospheric chemistry model simulations with the EMEP Chemistry Transport Model[60]. Using perturbation simulations, atmospheric transfer coefficients have been developed to relate emissions of $PM_{2.5}$ and its precursor gases NOx, $SO_2$, $NH_3$, and non-methane VOC to ambient $PM_{2.5}$ concentrations. For $PM_{2.5}$, NOx and $SO_2$, the coefficients explicitly distinguish contributions from urban and rural low-level sources such as MSW combustion.

**Description of the scenarios**. Socio-economic scenarios are important tools to assess the impacts of human activity on the environment and allow to explore alternative responses to mitigate and adapt to future alternative world developments[61]. A large set of scenarios have been developed[62,63], e.g., Global Scenario Group, IPCC-SRES, IPCC-TAR/AR4, UNEP GEO3/GEO4, OECD Environmental Outlook. We selected the Shared Socio-economic Pathways (SSPs)[47] as basis of our analysis because they are an important input for the recent and ongoing IPCC Assessment Reports and are central to the climate research community. An additional scenario, namely, IEA WEO 2018[34] has also been adopted to develop the ECLIPSE_V6b_Scenario. The baseline scenarios associated with the six socio-economic pathways describe the expected developments of MSW generation and management systems under current legislation 'CLE', hereafter baseline, i.e., assuming no further policies affecting the MSW sector are adopted until 2050. In addition, for each baseline an alternative scenario is constructed, which considers full implementation of circular MSW management systems globally and is referred to as the maximum technically feasible reduction 'MFR' scenario, hereafter mitigation scenario. Note that the technical frontier is explored here without taking account of the cost to implement various waste management strategies.

The MFR scenario is developed according to the SSP narratives and assumes a maximum technically feasible phase-in of a waste management system that is fully consistent with the EU's waste management hierarchy (Directive 2008/98/EC)[6]. This means that a first priority is given to technologies that circulate materials,

thereafter to technologies that recover energy, and only as a last resort to well-managed landfills. The following maximum recycling potentials of waste streams are applied: 90% of municipal paper and textile waste and 80% of municipal plastic and wood waste can be recycled. It is further assumed that 100% of food waste can be source separated and treated in anaerobic digesters with biogas recovery. These MFR potentials are adopted in consonance with the socioeconomic development for each scenario. Supplementary Information section 10 presents a description of the MFR management narratives specified for each scenario along with the regional aggregation.

**Uncertainty and limitations**. Regarding uncertainty, several data inputs (activity data, emission factors, type of management) go into the estimations, and therefore is difficult to do a quantitative uncertainty estimation[3,20]. Historical estimates of MSW generation, collection, management, and related emissions have associated uncertainties resulting from the different definitions of MSW coupled with contradictory reported values for generation and composition. The quality of the data suffers from inconsistencies in the definition of MSW generation across countries[52]. In some cases, amounts reported for MSW generation correspond to the gross quantities of waste collected and in other cases to the MSW quantities left for landfill after quantities separated for treatment have been deducted[51]. In developed countries, in particular, in Europe, MSW covers household waste and waste that is similar in nature and composition. In developing countries, data on waste suffers from incomplete characterizations and clear definitions of the fractions and source sectors included in the MSW are often lacking. These uncertainties are relatively high in developing countries compared to developed countries as in various cases data availability is quite limited in the former case[3]. In addition, some data reported for generation and collection refers to urban areas rather than national totals[4,53], which makes necessary to adopt assumptions based on dedicate studies for particular regions and expert knowledge to arrive at reasonable national MSW generation rates and attributions to urban and rural waste amounts. These uncertainties become bigger when estimating fractions of MSW openly burned as this information is in most of the cases not attainable. Furthermore, our study does not account for MSW trade nor for MSW generated from tourism. Therefore, estimates on unmanaged MSW and related GHG and air pollutants can be underestimated or on the contrary overestimated in some countries. Specific information on MSW composition for urban and rural areas is scarce. Projections on MSW composition are just an indication of future streams for the different settings and must be used with caution.

$CH_4$ emission factors are based on the IPCC Guidelines 2006[24], thereby carry out the uncertainties there described. Emissions factors for air pollutants and particulate matter depend on the composition of waste and burning conditions. Although we adopted the most recognized emission factors in the scientific arena, we acknowledge that large uncertainties are related to the values (uncertainties can be seen in ref. [20]) as those are estimated for total MSW and not specified by MSW fraction. Concerning uncertainty in projections, this is by some means assessed by adopting alternative activity scenarios which allow the comparison of the different estimates and reflect the sensitivities of the proposed measures to input assumptions[64].

Furthermore, the emission saving from energy recovery is not considered at this stage. The reason for this is that the fuel mix and the corresponding emission factors are important when assessing emission savings from energy recovery. This would mean that we need to explore the fuel mix and emission factors for every

scenario and region/country to be able to carry out a consistent analysis and provide robust results.

In general, there is a global need to improve information on MSW generation rates, treatment, and level of policy implementation[3]. Regardless of the uncertainties, we demonstrate the importance of improving global estimates of GHGs and air pollutant emissions from MSW and highlight the considerable role of this sector when assessing the respective mitigation potentials.

**Reporting summary**. Further information on research design is available in the Nature Research Reporting Summary linked to this article.

## Data availability

All data generated during this study is included in this published article (and its Supplementary Information). The Supplementary Data generated in this study has been deposited in http://pure.iiasa.ac.at/id/eprint/17598/[41]. Source data are provided with this paper.

**Databases used**

EDGAR v4.3.2. Gridded emissions of air pollutants for the period 1970-2012. https://essd.copernicus.org/preprints/essd-2017-79/

Environmental Protection Ireland. National waste statistics. https://www.epa.ie/our-services/monitoring--assessment/waste/national-waste-statistics/

Eurostat database. Table [env_wasmun] and Table [EU-SILC survey [ilc_lvho01]]. https://ec.europa.eu/eurostat/web/main/data/database.

OECD data. Table [Municipal Waste]. https://data.oecd.org/waste/municipal-waste.htm,

SSP Public Database version 2.0. Population, GDP and Urbanization data. https://tntcat.iiasa.ac.at/SspDb/dsd?Action=htmlpage&page=about,

The World Bank, 2019. What A Waste Global Database. World Bank Data Catalog. https://datacatalog.worldbank.org/search/dataset/0039597

UNDESA, 2018. World Urbanization prospects. https://population.un.org/wup/

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

## Acknowledgements

The development of the ECLIPSE_V6b scenarios was supported by the European Union funded Action on Black Carbon in the Artic ZK. The GEIGC Science and Technology Project on "Modelling Air Pollution Control and Environmental Health Perspectives under the Green and Low Carbon Transition of Global Energy System" (2900/2020-75001B) ZK.

## Author contributions

A.G.S. designed the study, performed all data analysis, and prepared the manuscript. G.K. performed the ambient air pollution concentration calculations. Z.K., W.S., and H.H. supervised the study. A.G.S., G.K., Z.K., W.S., and H.H. contributed to writing and revising the manuscript.

## Competing interests

The authors declare no competing interests.
