## [Peer Review File · Nature Communications]

REVIEWER COMMENTS

Reviewer #1 (Remarks to the Author):

Scenarios analysis are conducted by the authors where the MSW generation (globally) is projected, and different interventions are assumed to assess the emission reduction potential. The assessed issue/problem is valid and important.

However, I found that the structure and write up is not easy to follow. It is informative at the same time, lack of focus where the messages are dispersed and fail to deliver effectively. More effort is needed in improving this aspect.

The discussion and "scenario" link to SDGs is rather weak/artificial. The conclusion "The 6.3 target of the SDG 6 can only be achieved through more ambitious sustainability-oriented scenarios" is an obvious statement and say nothing. Some of the information in the results session appears to be more like a "method."

In term of the research, I have a few questions and would like the authors to clarify.

- Have you accounted and how do you consider the different filtration system in different countries? which could impact the net emissions
- Is the emission saving from energy recovery etc., in different countries being considered?
- Why do the authors choose to cluster EU by east and west?
- Figure 2 needs some improvement. Which region is having the strongest growth etc.? Good to indicate
- It is not too clear that what are the predictors in forecasting the MSW generation.
- How is MSW being defined in your study? Are they the same for all the countries? Are they separated? or unseparated?

Reviewer #2 (Remarks to the Author):

This is in interesting and important paper. As the authors write, their is a lack of global scenarios for future waste management. I therefore think and hope that the paper should be published eventually.

However, there is something I don't understand about the results. For example, in Table S1, results for CO2 emissions are presented and they are for some scenarios and years 0.0. I can't understand how they can be that. For example, incineration of plastics will generate CO2-emissions. Even if the recycling is increased, there will still be some incineration, so it can't be zero. Or is it so, that these emissions are for open burning only? But if so, where are the results for the total emissions? The paper is according to the title about global GHG emissions from waste management systems, so they should be somewhere I guess. In order to better understand these results I tried to find emission factors for different waste management methods of different waste fractions. However, I couldn't. Maybe there is a reference somewhere, but I couldn't find it. So I think the paper needs to be revised in order to better explain the results and provide data so that reviewers and readers can check the results. I would for example like to be able to see how much of each waste fraction is going to which treatment and what are the emission factors for these treatment methods for different years.

Reviewer #3 (Remarks to the Author):

Recommendation: Moderate revision.

This paper presents an interesting study, and its quality is excellent. Therefore, I think it could be suitable for publication after the issues raised above have been addressed.

1. Introduction

Q1. This section is a bit long and wordy. It is necessary to be more concise with a major focus on the problem investigated.

Q2. A list of abbreviations would be helpful to the future readers of the manuscript.

Q3. Line 62: the EU Waste Framework Directive has been updated. Please consider including some of these references:

- EU, 2018a. Directive 851 of the European Parliament and of the council of 30 May 2018 amending Directive 2008/98/EC on waste.

- EU, 2018b. Directive 852 of the European Parliament and of the Council of 30 May 2018 amending Directive 94/62/EC on Packaging and Packaging Waste.

- EU, 2018c. EU Directive 850 of the European Parliament and of the council of 30 May 2018 amending Directive 1999/31/EC on the landfill of waste.

Q4. Line 64: Regarding the MSW generation in high-income countries, I think this statement needs additional explanation. There are relevant differences between per capita generation and total generation. Other more recent studies than Ref. 9, such as:

<https://doi.org/10.1016/j.scitotenv.2021.147081>;

<https://doi.org/10.1016/j.scitotenv.2019.02.313>

shows that high-income European countries are reducing their generation per capita in the last years. Even, it can be checked in Eurostat database for UK, Ireland, Belgium, Spain, Netherlands. Please add some comments about it.

Results

Q5. Line 181. I am not sure, but in my view, results in Fig 2 for EU28 are not consistent with the EU policies and targets; please comment on them.

Q6. Would it be possible to show some parts of Fig. 2 more understandably?. Maybe EU-28 and India.

Conclusions

Q7, Some conclusions regarding the differences between high and low incomes countries could be added.

Q8. The authors should extend the policy recommendations.

Methods

Q9. The unit measurements in some mathematical equations should be explained.

Q10. Line 546. Why do you select the shared socio-economic pathway scenarios?. Then, please, justify by a new paragraph why other scenarios (mentioning them) were omitted.

Q11. A new table summarising the main differences between those selected scenarios would be helpful.

Q12. Line 564. Some limitations of the study are missing. The author/s should expose them.

Supplementary Information

Citation Lipinski et al., 2013 is missing

Comments to the author	Response from the author
Reviewer #1	
Scenarios analysis are conducted by the authors where the MSW generation (globally) is projected, and different interventions are assumed to assess the emission reduction potential. The assessed issue/problem is valid and important.	Thank you for this comment. We aimed to generate added value for research.
However, I found that the structure and write up is not easy to follow. It is informative at the same time, lack of focus where the messages are dispersed and fail to deliver effectively. More effort is needed in improving this aspect.	Thank you for this suggestion. We have revised the manuscript, thereby aiming to reduce jargon and make the text more readable throughout.
The discussion and "scenario" link to SDGs is rather weak/artificial. The conclusion "The 6.3 target of the SDG 6 can only be achieved through more ambitious sustainability-oriented scenarios" is an obvious statement and say nothing.	Thanks for this comment. We believe that is important to highlight that more ambitious targets and actions are urgently needed to reduce MSW generation and to globally adopt circular MSW systems to curb GHG and air pollutant emissions from this sector. We have modified this part of the discussion : However, more ambitious sustainability-oriented scenarios are crucial to meet the waste related SDGs, specially the 6.3 target which aims at “By 2030, improve water quality by reducing pollution, eliminating dumping and minimizing release of hazardous chemicals and materials, halving the proportion of untreated wastewater and substantially increasing recycling and safe reuse globally”⁴¹. We assess that under the current SSP1_MFR, it will not be possible to eliminate littering and open burning of MSW by 2030. Under this scenario the objective will be reached five years later i.e., in the year 2035. More ambitious targets and actions are urgently needed to reduce MSW generation and to globally adopt circular MSW systems and achieve additional GHG and air pollutant reductions Additionally, the global improvement of MSW systems has the potential to bring progress in other SDGs such as reduction food waste (SDG 2), avoiding the release of materials and plastics to water courses (SDG 14), access to energy through energy generation from anaerobic digestion and incineration (SDG 7), reducing GHG emissions to combat climate change and its impacts (SDG 13).

Some of the information in the results session appears to be more like a "method."	Thanks for noticing this. We have carried out the corresponding adjustments.
In term of the research, I have a few questions and would like the authors to clarify. -Have you accounted and how do you consider the different filtration system in different countries? which could impact the net emissions	Yes, we have accounted and considered the different filtration systems in different countries and even within a country. The filtration of the systems is looked up from two angles. One is the filtration of the MSW management systems and the other the filtration of technologies to remove air pollutants. The pace of adoption of different MSW technologies and air pollution controls is determined by the underlying narratives of the Shared Socioeconomic Pathways (SSPs) (O'Neill et al., 2017) and air pollution storylines (Rao et al., 2017). For example, in countries like China we differentiate the uptake of MSW technologies between Beijing, Shanghai and Hong Kong and rest of China. For India, we consider that the uptake of the MSW systems is different in Delhi. Similar filtration of the systems is assumed for all countries in LCAM. Furthermore, the composition of MSW is one of the most important aspects when quantifying emissions from MSW. We model the variability of MSW composition through the years which determines the technology adoption. Furthermore, filtration of technologies removing air pollutants are developed according to Rao et al 2017 where air pollution storylines are described. For instance, SSP5 has much more efficient air pollution controls than SSP3.
-Is the emission saving from energy recovery etc., in different countries being considered?	No, the emission saving from energy recovery is not considered at this stage. The reason for this is that the fuel mix and the corresponding emission factors are important when assessing emission savings from energy recovery. This would mean that we need to explore the fuel mix and emission factors for every scenario and region/country to be able to carry out a consistent analysis and provide robust results. We therefore believe that this aspect deserves a separate paper as a follow up to this paper. We have added this aspect as one of the limitations of the study.
-Why do the authors choose to cluster EU by east and west?	Thanks for your question. The reason behind is that EU west (Norway, Switzerland, and Iceland) have established MSW management systems in which recycling and incineration with energy recovery play a central role while in EU east countries, although transposing the EU waste related Directives into the national legislation, the development of the MSW management systems is still behind and MSW is mostly disposed of in landfills (managed and unmanaged). We therefore thought is important to expose these differences.
-Figure 2 needs some improvement. Which region is having the strongest growth etc.? Good to indicate	We have improved the explanation of Figure 2.

	Fig. 2. depicts MSW per capita in urban and rural areas for selected regions. NAM, Europe, Russia and Oceania are likely to continue having the highest average per capita MSW generation rates in both urban and rural areas during the whole period. However, the calculations reveal relevant differences between and within regions. By 2050, urban NAM is expected to generate in average 1008 (1017-1082) kg/cap/yr of MSW while rural NAM will generate 806 (814 – 864) kg/cap/yr of MSW. These MSW generation rates are about 35% higher than those estimated for the EU28 and 45% higher than those expected for Oceania. Our estimates also indicate that in a world following the SSP5, urban and rural China will generate between 31% and 36% more MSW per capita and year in 2050 than in 2015. The reason is the stronger economic growth projected in China over the next decade³⁵. In 2050, India is expected to generate between 16% and 25% less MSW per capita in urban and rural areas, respectively, compared to China for the same scenario. The lowest growth on MSW per capita in both urban and rural areas is expected in the SSP3 and SPP4. In general, Africa is projected to continue having the lowest MSW generation across scenarios during the whole period. An average of 355 (254 – 389) kg/cap/yr in urban areas and 155 (118-188) kg/cap/yr in rural areas of MSW is expected to arise in Africa in 2050. Supplementary Results. S1 displays total, urban, and rural waste generation by region and scenario.
-It is not too clear that what are the predictors in forecasting the MSW generation.	GDP per capita is the driver to project MSW generation. MSW generation per capita is projected on an estimated elasticity to GDP per capita for four different income groups. Furthermore, we have developed a methodology to explicitly estimate MSW generation in urban and rural areas. Supplement material section S6. presents a detailed description of the methodology to project MSW.
-How is MSW being defined in your study? Are they the same for all the countries? Are they separated? or unseparated?	Thanks for this question. We have clarified this aspect in the Methodology, section Municipal waste generation (MSW) activity and its characteristics. Here, we have primarily adopted the MSW definition stated in the Directive 851 of the European Parliament and of the council of 30 May 2018 amending Directive 2008/98/EC on waste: ‘Municipal waste is defined as waste from households and waste from other sources, such as retail, administration, education, health services, accommodation and food services, and other services and activities, which is similar in nature and composition to waste from households. Therefore, municipal waste includes, inter alia, waste from park and garden maintenance, such as leaves, grass and tree clippings, and

	waste from market and street cleaning services, such as the content of litter containers and sweepings except materials such as sand, rock, mud or dust’. However, the definition of MSW generation across countries suffers from inconsistencies thereby introducing high uncertainties to the estimates. In some cases, MSW generation amounts reported correspond to the gross quantities of waste collected and in other cases to the MSW quantities left for landfill after quantities separated for treatment have been deducted⁶². Furthermore, some countries include construction and demolition waste in the definition. In those cases, we have contrasted the maximum sources of information available to adapt the reported data to our core definition.
Reviewer #2	
This is an interesting and important paper. As the authors write, there is a lack of global scenarios for future waste management. I therefore think and hope that the paper should be published eventually.	Thank you for the encouraging feedback.
However, there is something I don't understand about the results. For example, in Table S1, results for CO₂ emissions are presented and they are for some scenarios and years 0.0. I can't understand how they can be that. For example, incineration of plastics will generate CO₂-emissions. Even if the recycling is increased, there will still be some incineration, so it can't be zero. Or is it so, that these emissions are for open burning only? But if so, where are the results for the total emissions? The paper is according to the title about global GHG emissions from waste management systems, so they should be somewhere I guess.	Thanks for raising this question. The emissions of CO₂ from incineration were indeed missing. We have now included the corresponding CO₂ emission estimations which are reflected in the manuscript, supplement, and graphs (excel file). We have included the following in the results section: Emissions of CO₂, particulate matter and air pollutants are released when MSW is burned. Our estimates suggest that globally 150 Tg/yr of CO₂ were emitted from MSW combustion in 2015. Please note that we only consider CO₂ emissions from fossil carbon, while CO₂ emissions from biogenic sources are considered net-zero. Our model indicates that NAM is the largest regional emitter of CO₂ from MSW (28 Tg/yr CO₂), followed by SASIA (24 Tg/yr CO₂), Oceania (19 Tg/yr CO₂), Africa (19 Tg/yr CO₂) and China (18 Tg/yr CO₂). However, the source of these emissions is different. While in NAM and Oceania the main source of emissions is MSW incineration (high quality incineration with energy recovery), in the remaining countries emissions are primarily generated from open burning of MSW. Under the current conditions, future CO₂ emissions in the baseline will increase proportionally to the quantities of MSW being incinerated and openly burned, resulting in an average emission of 263 Tg/yr CO₂ (242 – 308 Tg/yr CO₂) across SSPs in 2050 (Fig.5). Recycling of MSW plays a central part when diverting MSW

from combustion processes. The adoption of this measure together with the reduction of plastic MSW generation in the SSP1_MFR results in a maximum reduction of CO₂ emissions by 26% in 2050 compared to the corresponding baseline. All other MFRs scenario families bring CO₂ emission increases in the range from 20% to 25%.

We have updated Fig 5. in the main text – including the following sub-figure.

Fig. 5: Global amounts of MSW incinerated and burned in open fires and related emissions under CLE and MFR scenarios. Reduction fractions of MSW open burned result in the same reduction percentage of particulate matter and air pollutants. An increase of MSW incinerated with energy recovery is expected in all MFR scenarios. The reduction of CO₂ emissions results from the decline of MSW openly burned. Growth in CO₂ emissions in the MFRs families after 2030 is driven by the increase of MSW generation. The additional decline in CO₂ emissions in the SSP1_MFR is the consequence of reducing plastic waste generation. Supplementary Results S2 presents a detailed analysis of the MFR scenarios.

We have also updated the supplementary information.

In order to better understand these results I tried to find emission factors for different waste management methods of different waste fractions. However, I couldn't. Maybe there is a reference somewhere, but I couldn't find it. So I think the paper needs to be revised in order to better explain the results

Thanks for this comment. We have now included the emission factors for different waste management types in the supplementary information (excel file).

and provide data so that reviewers and readers can check the results.	
I would for example like to be able to see how much of each waste fraction is going to which treatment and what are the emission factors for these treatment methods for different years.	Emission factors along with waste flows by regions, stream and treatment have been included in the supplementary information (excel file).
Reviewer #3	
This paper presents an interesting study, and its quality is excellent. Therefore, I think it could be suitable for publication after the issues raised above have been addressed.	Thank you for the encouraging feedback.
1. Introduction Q1. This section is a bit long and wordy. It is necessary to be more concise with a major focus on the problem investigated.	Thanks for your suggestion. We have worked on improving the text.
Q2. A list of abbreviations would be helpful to the future readers of the manuscript.	Thank you for this suggestion. We have included a list of abbreviations in the Supplement: S3
Q3. Line 62: the EU Waste Framework Directive has been updated. Please consider including some of these references: - EU, 2018a. Directive 851 of the European Parliament and of the council of 30 May 2018 amending Directive 2008/98/EC on waste. - EU, 2018b. Directive 852 of the European Parliament and of the Council of 30 May 2018 amending Directive 94/62/EC on Packaging and Packaging Waste. - EU, 2018c. EU Directive 850 of the European Parliament and of the council of 30 May 2018 amending Directive 1999/31/EC on the landfill of waste.	Thanks for suggesting this. We have consulted and considered the different EU Directives. The references have been now included in the manuscript. Now it reads as follow: High-income countries can deploy policies and instruments to cope with the rising MSW flows and hence have cleaner and better-organized waste management systems. Examples include the EU Waste Framework Directive 2008/98/EC ⁶ and the amendment EU Directive 2018/851 ⁷ , the EU Landfill Directive 1999/31/EC ⁸ and the amendment EU Directive 2018/850 ⁹ , the EU Directive on packaging and packaging waste 94/62/EC ¹⁰ and the amendment EU Directive 2018/852 ¹¹ , the 3R's strategy in Japan ¹² and the Resource Conservation and Recovery Act 1976 ¹³ , 1986 in the United States.
Q4. Line 64: Regarding the MSW generation in high-income countries, I think this statement needs additional explanation. There are relevant differences between per capita generation and total generation. Other more recent studies than Ref. 9, such as:	Thanks a lot for providing us with very useful studies. We have consulted the references and we have modified the statement as follows: However, measures focusing solely on increasing re-use and recycling have a marginal impact on reducing waste generation ¹⁴ . Although some countries e.g., Japan and Netherlands, have managed to reduce MSW

https://doi.org/10.1016/j.scitotenv.2021.147081; https://doi.org/10.1016/j.scitotenv.2019.02.313 shows that high-income European countries are reducing their generation per capita in the last years. Even, it can be checked in Eurostat database for UK, Ireland, Belgium, Spain, Netherlands. Please add some comments about it.	generation, most of them are still not successful in reducing the per capita amounts of MSW generated each year¹⁵ Eurostat shows that some high-income countries have reduced per capita MSW generation for some periods of time. However, this reduction has not been maintained in all countries and recently an increase on per capita and total MSW is observed (see graphs below using Eurostat data Table [env_wasmun]). We have highlighted relevant differences between per capita and total generation in the Results. Specifically, we have included the following: North America (NAM), Europe, Russia and Oceania are likely to continue having the highest average per capita MSW generation rates in both urban and rural areas during the whole period. However, the calculations reveal relevant differences between and within regions. By 2050, urban NAM is expected to generate in average 1008 (1017-1082) kg/cap/yr of MSW while rural NAM will generate 806 (814 – 864) kg/cap/yr of MSW. These MSW generation rates are about 35% higher than those estimated for the EU28 and 45% higher than those expected for Oceania. Our estimates also indicate that in a world following the SSP5, urban and rural China will generate between 31% and 36% more MSW per capita and year in 2050 than in 2015. The reason is the stronger economic growth projected in China over the next decade³⁵. In 2050, India is expected to generate between 16% and 25% less MSW per capita in urban and rural areas, respectively, compared to China for the same scenario. The lowest growth on MSW per capita in both urban and rural areas is expected in the SSP3 and SPP4. In general, Africa is projected to continue having the lowest MSW generation across scenarios during the whole period. An average of 355 (254 – 389) kg/cap/yr in urban areas and 155 (118-188) kg/cap/yr in rural areas of MSW is expected to arise in Africa in 2050.
Results Q5. Line 181. I am not sure, but in my view, results in Fig 2 for EU28 are not consistent with the EU policies and targets; please comment on them.	Thanks for this comment. In our interpretation, the Directive 851 of the European Parliament and of the council of 30 May 2018 amending Directive 2008/98/EC on waste Article 9 does not represent an official target for waste reduction and therefore we did not assume any specific policies regarding MSW reduction when forecasting MSW generation for the EU28 in any of the baseline scenarios. Extracted from the Directive: “6. By 31 December 2023, the Commission shall examine the data on food waste provided by Member States in accordance with Article 37(3) with

	a view to considering the feasibility of establishing a Union-wide food waste reduction target to be met by 2030 on the basis of the data reported by Member States in accordance with the common methodology established pursuant to paragraph 8 of this Article. To that end, the Commission shall submit a report to the European Parliament and to the Council, accompanied, if appropriate, by a legislative proposal.” In terms of treatment/management we have considered the following directives. Directive 851 of the European Parliament and of the council of 30 May 2018 amending Directive 2008/98/EC on waste. Directive 852 of the European Parliament and of the Council of 30 May 2018 amending Directive 94/62/EC on Packaging and Packaging Waste. Directive 850 of the European Parliament and of the Council of 30 May 2018 amending Directive 1999/31/EC on landfill waste.
Q6. Would it be possible to show some parts of Fig. 2 more understandably? Maybe EU-28 and India.	We have modified the scales for all regions in Fig.2. They are more understandable now. It is easier to identify the different MSW trajectories for the regions shown.
Conclusions Q7, Some conclusions regarding the differences between high and low incomes countries could be added.	Thanks for this suggestion. We have included the following paragraph: Significant potentials exist to reduce GHG, and air pollution provided the implementation of circular MSW management systems. However, the maximum reduction potentials differ between and within regions. Different scenario developments result in similar reduction trajectories in urban and rural areas in high-income regions. However, in low-and middle-income regions different developments lead to different emission trajectories. Rural areas will be drastically affected in scenarios in which inequalities prevail.
Q8. The authors should extend the policy recommendations.	In addition to policies related to increasing collection rates and diverting waste from landfills/dumpsites and open burning through reducing and recycling, we have added the following: Certainly, the adoption of circular MSW systems will require actions that incentivize the market of secondary materials and induce consumers and producers to make use of these resources. Regulatory policies associated to assure circularity of products are also necessary. However, its essential to adopt global binding measures to reduce MSW generation to guarantee the success.

Methods Q9. The unit measurements in some mathematical equations should be explained.	Thanks for noticing this. We have included the units in the equations.
Q10. Line 546. Why do you select the shared socio-economic pathway scenarios? Then, please, justify by a new paragraph why other scenarios (mentioning them) were omitted.	Socio-economic scenarios are important tools to assess the impacts of human activity on the environment that allow researchers to explore alternative responses to mitigate and adapt to future alternative world developments⁵⁹. A large set of scenarios have been developed over the last years^{60,61} e.g., Global Scenario Group, IPCC-SRES, IPCC-TAR/AR4, UNEP GEO3/GEO4, OECD Environmental Outlook. We selected the Shared Socio-economic Pathways (SSPs)⁴⁷ as basis of our analysis because they are an important input for the recent and ongoing IPCC Assessment Reports and are central to the climate research community. An additional scenario, namely, IEA WEO 2018³⁴ has also been adopted to develop the ECLIPSE_V6b_scenario. We have included that in the Methods section
Q11. A new table summarizing the main differences between those selected scenarios would be helpful.	A new table summarizing the main differences between the scenarios is included in the Supplement Methods Table S2.
Q12. Line 564. Some limitations of the study are missing. The author/s should expose them.	We have renamed the section ‘Uncertainties and limitations.’ In addition to what we already had we added the following: Furthermore, our study does not account for MSW trade nor for MSW generated from tourists. Therefore, estimates on unmanaged MSW and related GHG and air pollutants can be underestimated or on the contrary overestimated in some countries. Specific information on MSW composition for urban and rural areas is scarce. Projections on MSW composition are just an indication of future streams for the different settings and must be used with caution. Also added the last part in this sentence: Although we adopted the most recognized emission factors in the scientific arena, we acknowledge that large uncertainties are related to the values (uncertainties can be seen in ref²⁰) as those are estimated for total MSW and not specified by MSW fraction.
Supplementary Information	Thanks for noticing this. We have added the corresponding reference.

Citation Lipinski et al., 2013 is missing

Reviewer # 3: Q4:

REVIEWERS' COMMENTS

Reviewer #1 (Remarks to the Author):

I am happy with the revision and would like to recommend it for publication. The raised questions/comments have been well addressed where clarification is provided, including those related to the methodology.

Reviewer #2 (Remarks to the Author):

Thank you for this revised version. I think you have addressed the comments given in the first round very well. I have only some minor comments:

- * Line 74 and 75. This statement cannot be true. Please check it.
- * Fig 4 and 5. I don't understand how emissions and waste amounts can be negative. Please revise or explain better what the figures are showing.
- * Fig 6. In the figure it says PM2.5 but the figure legend says BC. Please check.
- * Lines 358 to 360. I think this statement needs to be qualified in terms of probabilities and scenarios.
- * Line 495. Please check language.

Reviewer #3 (Remarks to the Author):

The authors answered in a courteous manner and the manuscript is corrected appropriately. Thank you for improving the quality of your work.

Comments to the author	Response from the author
Reviewer #1	
I am happy with the revision and would like to recommend it for publication. The raised questions/comments have been well addressed where clarification is provided, including those related to the methodology.	Thank you for this comment. We are happy to hear that we have addressed your comments correctly.
Reviewer #2	
Thank you for this revised version. I think you have addressed the comments given in the first round very well. I have only some minor comments:	Thanks a lot for your feedback.
Line 74 and 75. This statement cannot be true. Please check it.	We have modified the statement as follows: High-income countries can deploy policies and instruments to cope with the rising MSW flows and hence they can potentially have cleaner and better-organized waste management systems
Fig 4 and 5. I don't understand how emissions and waste amounts can be negative. Please revise or explain better what the figures are showing.	The negative values on the y-axis in Fig 4 and Fig 5 refer to avoided emissions (Fig 4) and prevented quantities of MSW that would otherwise be openly burned (Fig 5). For example, Fig 4 shows that in the SSP1_MFR, it would be possible to avoid the release of 4.6, 28.38 and 46.98 Tg CH₄/yr compared to its corresponding counterparts in the SSP1_CLE by 2030, 2040 and 2050, correspondingly. Fig 5 shows that in the SSP1_MFR, it would be possible to prevent the open burning of 546, 680 and 779 Tg MSW/yr compared to its corresponding counterparts in the SSP1_CLE by 2030, 2040 and 2050, respectively. We have added the following in the caption of the figures: Fig 4. Negative values in y-axis in panel c refer to avoided emissions. Fig 5. Negative values in y-axis in panel c refer to avoided open burning of MSW.
Fig 6. In the figure it says PM2.5 but the figure legend says BC. Please check.	Thanks for noticing this. We have carried out the corresponding amendment.
Lines 358 to 360. I think this statement needs to be qualified in terms of probabilities and scenarios.	We have modified the statement as follows: Under the current conditions, future CO ₂ emissions in the baseline would probably increase proportionally to the quantities of MSW being incinerated and openly burned, resulting in an average emission of 263

	Tg/yr CO ₂ (in the range from 242 in SSP3 to 308 Tg/yr CO₂ in SSP5) across SSPs in 2050 (Fig.5). Recycling of MSW plays a central part when diverting MSW
Line 495. Please check language	Thanks for highlighting this. We have modified the sentence as follows: Rural waste generation is estimated by applying shares representing the relationship between urban and rural waste generation per capita for different regions.
Reviewer #3	
The authors answered in a courteous manner and the manuscript is corrected appropriately. Thank you for improving the quality of your work.	Thank you for this comment. We are pleased to hear that we have addressed your comments correctly which help us to improve our work.